# Degree of Hydrolysis Regulated by Enzyme Mediation of Wheat Gluten Fibrillation: Structural Characterization and Analysis of the Mechanism of Action

**DOI:** 10.3390/ijms241713529

**Published:** 2023-08-31

**Authors:** Huijuan Zhang, Shihao Lv, Feiyue Ren, Jie Liu, Jing Wang

**Affiliations:** 1China-Canada Joint Lab of Food Nutrition and Health (Beijing), Beijing 100048, China; 2Key Laboratory of Special Food Supervision Technology for State Market Regulation, Beijing 100048, China; 3School of Food and Health, Beijing Technology & Business University (BTBU), Beijing 100048, China; lvsh0927@163.com (S.L.); 20200101@btbu.edu.cn (F.R.); liu_jie@btbu.edu.cn (J.L.)

**Keywords:** wheat gluten, amyloid fibrils, wheat gluten peptides, trypsin, degree of hydrolysis

## Abstract

The impact of different degrees of hydrolysis (DHs) on fibrillation when trypsin mediates wheat gluten (WG) fibrillation has not been thoroughly investigated. This study discussed the differences in amyloid fibrils (AFs) formed from wheat gluten peptides (WGPs) at various DH values. The results from Thioflavin T (ThT) fluorescence analysis indicated that WGPs with DH6 were able to form the most AFs. Changes in Fourier Transform Infrared (FTIR) absorption spectra and secondary structure also suggested a higher degree of fibrillation in DH6 WGPs. Analysis of surface hydrophobicity and ζ-potential showed that DH6 AFs had the highest surface hydrophobicity and the most stable water solutions. Scanning Electron Microscopy (SEM) and Transmission Electron Microscopy (TEM) images revealed the best overall morphology of DH6 AFs. These findings can offer valuable insights into the development of a standardized method for preparing wheat gluten amyloid fibrils.

## 1. Introduction

Amyloid fibrils (AFs) have gained significant attention due to their remarkable properties, including a wide range of raw materials, high aspect ratio, excellent stiffness, and abundant functional groups on the surface [1]. The exceptional properties of AFs originate from their characteristic structural features, known as cross-β structures [2]. A cross-β structure comprises two or more β-sheets parallel or anti-parallel superimposed at a specific distance [3]. Multiple cross-β structures are superimposed to form a cross-β superstructure, multiple super-cross-β structures connect in a parallel arrangement to form a protofilament through both covalent and non-covalent bonds, and several protofilaments twist together to form a mature AF [1,4]. Compared to other protein aggregates, the cross-β structures, specifically the sterically zipper-ridged structure, provide AFs with exceptional stability [3]. AFs can be produced from a diverse range of raw materials, including animal and plant proteins under appropriate conditions [5,6]. Utilizing plant proteins instead of animal proteins for the production of AFs is particularly appealing to minimize carbon emissions.

WG is one of the vital plant proteins in daily consumption [7]. It is a by-product of wheat starch in industrial separation. Although it has been widely used in food preparation and animal feed, other applications for this plant protein are currently being explored [8]. Fibrillating wheat gluten proteins is an interesting and promising strategy for enriching protein function. Wheat gluten is basically composed of 40–50% gliadin and 30–40% glutenin [9,10]. Unlike other proteins, WG is rich in glutamine repeats and contains proline at the end of the sequence [4]. This is similar to the “Q-block” that can easily form AFs, as mentioned by Chen et al. [11]. WG also contains many hydrophobic amino acids, which can provide a large number of hydrophobic bonds for cross-β structure formation [12]. The large number of β-sheet structures contained in WG also provides a quantitative basis for the formation of cross-β structures [12,13]. There are many proteins that can form AFs, such as egg white lysozyme [14], β-Lactoglobulin [15,16], insulin [17], soy protein [18], rice protein [19], oat protein [20], etc. The key to the formation of AFs is unfolding, followed by self-assembly [1]. However, almost all of the above proteins are unfolded under extreme conditions such as strong acids, whereas insoluble WG can be hydrolyzed by trypsin into peptide solutions for unfolding purposes, and then produce AFs under heated conditions, avoiding a strong acidic environment [4,21].

Currently, the majority of research on wheat gluten amyloid fibrils focuses on investigating the conditions under which fibrillation occurs. Athamneh et al. conducted a study on the effects of tryptic hydrolysis at pH 5.7 and pH 8 on fibrillation [4]. Ridgley et al. utilized AFM and SEM to depict the four-stage process of wheat gluten amyloid fibril formation [22]. Ridgley and Claunch et al. proposed a hypothesis known as the “template” theory mechanism for the formation of wheat gluten amyloid fibrils, where tryptic hydrolysis products of gliadin act as “template” proteins and glutenin hydrolysis products act as “adder” proteins [12]. Furthermore, the size, shape, and stiffness of AFs can be influenced by various processing conditions such as temperature, pH, and ionic strength [23]. Lambrecht et al. studied the impact of different enzyme treatments, including trypsin, proteinase K, papain, chymotrypsin, and thermolysin, on the fibrillation of WG [24]. In the preparation of AFs, the biggest difference between WG and other proteins was the use of trypsin to hydrolyze the proteins for the purpose of unfolding. Therefore, it is crucial to study the effect of the DH of WG on fibrillation.

Based on several previous studies, the aim of this study was to investigate the effect of different DHs on WG fibrillation. Using surface hydrophobicity analysis, ζ-potential analysis, FTIR absorption spectroscopy analysis, secondary structure analysis, and gel electrophoresis, the differences among WGPs with different DHs and the effect on AF formation was investigated. AFs were identified by ThT fluorescent assay and FTIR, and morphologically characterized by SEM and TEM.

## 2. Results

### 2.1. Optimal Enzyme–Substrate Ratio

The water insolubility of WG largely limits its development and application [25], but the use of trypsin to hydrolyze WG can effectively solve this problem [26]. The approximate enzymatic time and optimal enzyme–substrate ratios for different DHs can be derived from Table 1. As the DH continued to deepen, the enzymatic time increased significantly. The DH is the number of hydrolyzed peptide bonds as a percentage of the total number of peptide bonds per unit weight in WG [27]. The higher the degree of hydrolysis, the more peptide bonds are hydrolyzed. Therefore, under the same enzyme–substrate ratio, the higher the degree of hydrolysis, the higher the enzymatic hydrolysis time required. There was no significant difference between the enzymatic times for different DH enzyme–substrate ratios of 1:100 and 1:500, but the enzymatic time for 1:1000 was significantly higher than both. The findings indicated that the higher the DH, the more enzymatic time was required. When the enzyme–substrate ratio was 1:1000, the enzyme was saturated by the substrate, and increasing the enzyme concentration at this point sped up the enzymatic reaction rate. Enzymatic time was significantly lower than 1:1000 when the enzyme–substrate ratio was 1:500. However, when the enzyme concentration was increased to 1:100, there was no significant increase in the enzymatic time, which may be due to the fact that the saturating concentration of the enzyme was already reached at the enzyme–substrate ratio of 1:500. Therefore, the optimal ratio of enzyme to substrate was found to be 1:500.

### 2.2. Analysis SDS-PAGE

The results of gel electrophoresis molecular weight distribution of DH4–8 WGPs and DH4–8 AFs are shown in Figure 1. Figure 1A shows that after WG was hydrolyzed by trypsin, the insoluble WG was partially hydrolyzed into smaller molecules of peptides ranging 11~17 kDa, and partially hydrolyzed into peptides of 35 kDa. The study by Athamneh et al. also confirmed that trypsin hydrolysis produced smaller peptides [4,28]. Fibrillation of WGPs with different DHs resulted in significantly lighter bands at 11~17 kDa. The color of the band at 245~kDa was significantly deeper as shown in Figure 1B, indicating that these small-molecule peptides were converted to large molecules by fibrillation. Combined with the following ThT fluorescence assay, FTIR absorption spectra, SEM, and other analytical tools indicate that these macromolecules were AFs.

### 2.3. ThT Fluorescence Intensity

WGP fibrillation was assessed by measuring ThT fluorescence intensity [29] based on previous studies showing that the enhancement of ThT fluorescence intensity indicates cross-β structure generation [18,24]. According to Figure 2, the ThT fluorescence intensity of DH4–8 WGPs increased to varying degrees from the beginning of the incubation at 0 h to the end of the incubation at 48 h. The WGPs of DH6 had the highest ThT fluorescence intensity after 36 h of incubation and was significantly higher than the other four DHs of the WGPs. Notably, after incubating for 36 h, the ThT fluorescence intensity essentially did not increase, indicating that AFs were no longer being generated in the WGP solutions of DH4–8. This result was supported by Lambrecht et al. [24]. It can be inferred that the “raw material”, β-sheet-rich peptides that support AF formation, may have been exhausted after 36 h of incubation of the WGP solutions. The maximum ThT fluorescence intensity of DH4 WGPs solutions were significantly lower than those of DH5–8; this may be due to the lower degree of hydrolysis leading to a lower degree of WG folding, releasing only a small amount of hydrophobic amino acids, and resulting in the driving force of AF formation—the hydrophobic force is low, unable to form a large number of AFs. In summary, the DH6 WGP solutions was able to form the most AFs.

### 2.4. Analysis FTIR Absorption Spectra

According to Figure 3, the FTIR absorption spectrum of WG and DH4–8 WGPs and AFs were detected in the range of 400–4000 cm^−1^. The bands from 1600 cm^−1^ to 1700 cm^−1^ can respond to changes in secondary structures associated with the amide I because the spectral bands in this region are highly sensitive to small changes in molecule geometry resulting from the C=O stretching vibration frequency [30], where the vibrational band of 3000–3800 cm^−1^ was related to the stretching modes of free O-H and hydrogen-bonded O-H [31]. Figure 3A–E show the WG to AF FTIR spectra of DH4, DH5, DH6, DH7, and DH8. Appendix A show the secondary structure fitting diagram of DH4–8 WGPs and AFs. The area of each peak represents the level of the secondary structure to which this region belongs. Table 2 shows the proportion of WGP secondary structures derived from the fitted treatments in Appendix A. Table 3 shows the proportion of AF secondary structures derived from the fitted treatments in Appendix A.

After WG was hydrolyzed and then fibrillated, the FTIR spectra of DH4–8 WGPs and AFs showed significantly different stretching vibrations at 3275–3284 cm^−1^, suggesting that the hydrogen bonding of AFs was affected by different DH. The peaks in the amide I region were all shifted to the left to varying degrees, indicating the generation of cross-β structures at different DHs. Trypsin hydrolysis of WG released short glutamine-rich peptides that were favorable for the formation of cross-β structures [4,11]. Combining the changes in secondary structure derived in Table 2 and Table 3, it can be inferred that some α-helical structures undergo an α-to-β transition on the surface of the cross-β structures. Ridgley et al., based on the spectral changes of the self-assembled peptide mixture (gliadin hydrolysate, glutenin, myoglobin hydrolysate, etc.), also showed that the content of the α-helix structure decreased and the content of the cross-β structure increased [13]. WG is basically composed of gliadin and glutenin, and its hydrolysates undergo a similar transformation (as shown in Table 2 and Table 3). This indicated that the α-to-β transition is crucial for the formation of the basic units before fibril formation. The absorption peak of DH6 AFs was shifted the furthest distance to the left, which indicated that DH6 had the highest degree of WGP fibrillation. This was consistent with the results exhibited by ThT fluorescence assay. DH4 AFs had the lowest number of shifted waves of absorption peaks, indicating a lower degree of fibrillation. This was attributed to the lower degree of hydrolysis, with more unhydrolyzed large peptides in solutions, which were able to form the conventional β-sheets, but not further AFs [4]. When the DH reached 6, there were enough short peptides with strong hydrophobicity in solutions to form cross-β structures and grow into mature AFs [24]. However, when further hydrolyzed to DH7 or even DH8, the degree of fibrillation was rather reduced. The rational explanation is that its reduced hydrophobicity led to an insufficient driving force for the formation of AFs. That is, there was not enough hydrophobic force for the α-helix to undergo an α-to-β transition at the surface of the cross-β structures [12]. This can also be demonstrated by the changes in secondary structure in Table 2 and Table 3: Δβ-sheets increase with increasing hydrolysis, but decrease when reaching DH7 and DH8.

### 2.5. Analysis of Surface Hydrophobicity and ζ-Potential

The nonpolar surfaces exposed in WGPs and AFs were examined using ANS, and the surface hydrophobicity results are shown in Figure 4A. The surface hydrophobicity of WGPs is the key to their fibrillation [24]. When trypsin hydrolyzes WG, it selectively cleaves only positively charged lysine and arginine, thus releasing peptides containing more hydrophobic amino acids [25]. From Figure 4A, it can be concluded that DH6 WGPs have the highest surface hydrophobicity. This may be because the surface hydrophobicity of the solutions increased as more peptides containing hydrophobic amino acids within the protein were exposed as hydrolysis deepened [32]. When DH7 and DH8 were reached, more highly hydrophobic peptides were present in the solutions, causing the system to become unstable. These shorter peptides were in turn bound by hydrophobic forces and hydrogen bonds to form large irregularly curled peptides, thus hiding the hydrophobic amino acids inside and reducing the surface hydrophobicity of the solutions [25]. In addition, hydrophobic interactions are an important property for the formation of cross-β structures [12]. Figure 4A shows that fibrillation obviously boosted the surface hydrophobicity of proteins, whereas the AFs generated by the WGPs of DH6 also had the highest surface hydrophobicity. A plausible explanation is that during the formation of the cross-β structures and their growth into AFs, hydrophobic amino acids were present in the side chains and exposed on the surface of the AFs, thus giving the AFs a high surface hydrophobicity [13]. This makes them excellent carriers of nutrients or drugs, enabling them to reach the body’s optimal absorption sites [33].

The ζ-potential is a critical parameter to characterize the solutions’ stability [34]. Figure 4B shows that both WGPs and AFs with different DHs were negatively charged at pH8. DH6 WGPs and AFs had the highest ζ-potential. Similar to surface hydrophobicity, more charged amino acids were exposed as hydrolysis deepened [35]. At DH7 or even DH8, the excess of short peptides after binding into large irregularly curled peptides through hydrophobic forces and hydrogen bonding resulted in the charged amino acids being hidden again inside the protein, which in turn reduced the ζ-potential. The ζ-potential value of WGPs with different DHs were significantly increased after fibrillation. This may be due to the fact that during fibrillation, proteins unfold, exposing more charged amino acids, which may lead to the formation of more stable systems. There have been reports that systems with more charged amino acids on the protein surface are more stable [36]. Hence, WGPs have inferior colloidal stability compared to AFs. This was verified by the findings of surface hydrophobicity. Furthermore, both WGPs and AFs derived from DH6 exhibit the highest ζ-potentials, indicating that DH6 is the most suitable level of hydrolysis for the production of AFs. DH6-based AFs possess the greatest colloidal stability. Additionally, electrostatic interaction plays a crucial role in fibrosis progression. Greater electrostatic interaction results in organized growth and increased stiffness of AFs [16]. This gives the DH6 AFs the highest stiffness as well.

### 2.6. Analysis of SEM

SEM was used to observe the surface morphology structure and diameter of AFs formed by WGPs at different degrees of hydrolysis (DH) [4,13]. The diameters of the AFs were determined using Image J (2.3.0) software. Figure 5 presents the SEM images of the DH4–8 AFs. The AFs generated by the DH4 WGPs exhibited an irregular shape with branches and a diameter of only 4.2 µm. Upon reaching DH5, the diameter of the AFs increased to 7 µm, but the thickness remained non-uniform. The WGPs of DH6 produced the AFs with the largest diameter, measuring up to 8.86 µm, with uniform thickness. However, as the degree of hydrolysis reached DH7 or DH8, the diameters of the AFs started to decrease to 4.1 µm and 3.3 µm, respectively. These findings indicate that thicker and straight AFs can be formed as the degree of hydrolysis of WGPs increases. The optimum degree of hydrolysis for WGPs was found to be DH6, as it enabled the formation of straight and thick AFs. Further hydrolysis led to a decrease in the diameter of the AFs formed by WGPs due to the weakening of hydrophobic forces and the decrease in dispersion stability.

### 2.7. Analysis of TEM

TEM can be used to study the overall morphology of AFs [21,24]. TEM images of the DH4–8 AFs are shown in Figure 6. The DH4 WGPs generated short worm-like AFs. This could be explained by the fact that the DH was too low, resulting in a lack of suitable short peptides to form cross-β structures and continue to grow into AFs. DH5 WGPs generated AFs improved, their length increased, and they were no longer curled together. Compared to DH4 WGPs, DH5 WGPs underwent a more extensive hydrolysis process, resulting in the release of a greater number of short peptides with hydrophobic properties. This enabled DH5 WGPs to effectively form cross-β structures and promote the normal growth of AFs. DH6 WGPs formed straight and the longest AFs among other degrees of hydrolysis. At this stage, the WGP solution contained a significant number of highly hydrophobic and stabilized peptides, resulting in the formation of AFs with the best morphology. This result was also consistent with the ThT fluorescence and FTIR findings. Although the DH7 WGPs also formed straight AFs, they were shorter in length. DH8 WGPs also formed straight AFs, but at a shorter length than DH7. One explanation for these observations is that excessively high levels of hydrolysis cause the peptides in the WGP solution to recombine and form large peptides or amorphous aggregates through non-covalent bonds, including hydrophobic forces and hydrogen bonds [37]. These large peptides or aggregates are unable to further contribute to the formation of AFs [1,25].

## 3. Discussion

### 3.1. Possible Mechanism Analysis of AFs’ Formation Process of WG

The fibrillation process of WG can be summarized as hydrolysis, nucleation, growth, and maturation. The ‘template’ theory hypothesis proposed by Ridgley et al. was a good fit for the explanation of the fibrillation process of WG [12,38]. Trypsin hydrolysis of WG released a high concentration of the ‘template’ peptide Gd20 from gliadin, one of the constituents of WG [38]. The ‘template’ peptides are highly hydrophobic, and they can bind to each other to form a β-sheet to stabilize themselves by hiding the hydrophobic groups between the sheets [22]. Multiple templates superimposed in this way result in a cross-β structure. However, this structure still has two exposed hydrophobic surfaces. The ‘adder’ peptides enzymatically released from glutenin, one of the constituents of WG, have a highly α helix [12]. When they contact the hydrophobic surface of the cross-β structures, the α helix will unfold on the surface and ‘add’ to the cross-β structure [22]. At this point a complete ‘nucleus’ is formed, which can also be called a cross-β superstructure [4]. Next, a large number of ‘nuclei’ grow in parallel to form a single protofilament, and then multiple protofilaments are twisted by non-covalent or covalent bonds to form a mature AF [1,4].

As the first step in fibrillation, the hydrolysis stage plays a very important role. It is the foundation of the entire fibrillation process. The mechanisms of different DH-regulated WG fibrillations are shown in Figure 7 (taking DH4, DH6, and DH8 as examples). When the enzymatic reaction reached DH4, a smaller amount of the WG was hydrolyzed, resulting in the release of only a limited number of ‘template’ peptides and ‘adder’ peptides. Consequently, only a small number of internal hydrophobic groups are exposed. This limited exposure leads to the formation of only a small number of cross-β structures and ‘nuclei’. Additionally, due to insufficient hydrophobic forces, these structures are unable to effectively grow into a mature AF [12]. Consequently, as shown by SEM and TEM images of DH4, the DH4 AF exhibits a worm-like morphology with a thin diameter. When the enzymatic reaction reached DH6, most of the WG was hydrolyzed and plentiful ‘template’ and ‘adder’ peptides were released. The higher degree of unfolding allowed a mass of internal hydrophobic groups to be exposed and plenty of ‘nuclei’ to form and grow into mature AFs [2]. As a result, the DH6 AF was long, straight, and thick in diameter. When the enzymatic reaction reached DH8, WG was essentially hydrolyzed, and a great deal of short peptides, template peptides, and additive peptides were present in the solution. The excessive exposure of hydrophobic groups results in a decrease in the stability of the solution. In order to stabilize themselves, certain peptides in the solution bind or aggregate with each other [1]. However, the current solution conditions are not ideal for fibrillation. As a result, most of the peptides form amorphous aggregates, while only a few of them undergo amyloid fibrillation in the subsequent stages. As a result, the DH8 AF was shorter than the DH6 AF and its diameter was thinner.

### 3.2. Concluding Remarks and Future Perspectives

An important direction to expand the application of wheat gluten is based on trypsin-mediated wheat gluten fibrillation. So far, only the optimal conditions in terms of pH, temperature, and protein concentration have been investigated, although there have been some studies using trypsin to enzymatically digest wheat gluten proteins to produce amyloid protofibrils. The effects of DH2 and DH6 on WG fibrillar formation were also reported by Lambrecht et al. [24]. Based on this, our study will be further extended to investigate the effect of degree of hydrolysis on WG fibril formation on a larger scale. Fluorescence detection, FTIR absorption spectroscopy, SEM, and TEM showed that the hydrolysis of WGs to DH6 could produce a higher number of AFs with better morphology, the highest surface hydrophobicity, and a more stable system under suitable conditions. These results further optimized the existing methods for the preparation of wheat gluten amyloid fibrils. These results further optimized the existing methods for the preparation of wheat gluten amyloid fibrils, provided valuable insights into the DH regulating WG fibrillation, and provided a basis for the application of wheat gluten amyloid fibrils. Moreover, this study provided a reference for the fibrillation of other insoluble proteins or proteins that are difficult to unfold. The regulation of DH is very important for the process of WG fibrillation. It can directly affect the raw materials and driving force for the formation of AFs, causing AFs to exhibit different amounts and morphologies. This resulted in different properties of the AFs formed by WGPs with different DHs. The determination of the optimal DH maximizes the formation of wheat gluten amyloid fibrils with excellent properties and improves industrial productivity.

However, the current formation of AFs from WG was still low in yield, slow, and lacked an effective method to be purified. Future research may need to focus on these shortcomings, such as the addition of a fibrillating agent or a special mechanical treatment, to further improve the yield and the rate of fibrillation of WG. Few studies have also been reported using gluten protein amyloid protofibrils, which may hold great promise for the future. Similarly, lysozyme AFs can form a double-networked hydrogel with ferulic acid and chitosan to achieve AGE inhibition [39]. Hydrogels with antimicrobial activity can be formed by combining EGCG and lysozyme AFs [40]. Whey protein AFs act as a carrier for curcumin, which improves the stability of its dispersion in water and enhances its antioxidant activity [33]. Thus, there is hope that wheat gluten amyloid fibrils will also contribute to the food, biology, and pharmaceutical sectors.

## 4. Materials and Methods

### 4.1. Materials

Wheat gluten [containing 68.4% protein, and an automatic Kjeldahl nitrogen analyzer (Kjeltec 8400) was used for determination], Trypsin from bovine pancreas (T8003), Thioflavin T (ThT) (D844310), ANS-Na (=Sodium 8-Anilino-1-naphthalenesulfonate) (A866743), and sodium hydroxide were all obtained from Macklin (Shanghai, China). All reagents were at least of analytical grade.

### 4.2. Hydrolysis of WG

WG proteins were hydrolyzed at 37 °C, 200 rpm. First, the WG suspension (2.5% *w*/*v*) was hydrolyzed to DH4, DH5, DH6, DH7, and DH8 using trypsin. All hydrolysis experiments were carried out in triplicate. All data represent the average of independent measurements. The enzyme–substrate ratios of each DH were 1:100, 1:500, and 1:500, respectively. WG was continuously hydrolyzed at 37 °C with a pH of 8. When the required DH was reached, Trypsin was inactivated at 97 °C, followed by cooling and centrifugation (9930 rpm, 15 min). The supernatant was taken and freeze-dried into wheat gluten peptide powder for subsequent use.

Second, the pH was measured every hour and the solution pH was maintained at 8 using 1 M NaOH during the hydrolysis process. The amount of sodium hydroxide used each time was recorded to calculate the DH. DH is defined as the number of hydrolyzed peptide bonds as a percentage of the total number of peptide bonds per unit weight in WG. The calculation formula is as follows:(1)$$DH=\frac{h}{h_{tot}}=\frac{B\cdot M_{b}\times 100}{\alpha \cdot m_{p}\cdot h_{tot}}$$
where *h* is the hydrolysis equivalents, *h_to_*_t_ is the theoretical number of peptide bonds present in the protein per unit weight, *B* is the total amount of base added, *M_b_* is the molarity of the base added, *α* is a measure of the degree of α-NH3+ dissociation, *m_p_* is the mass of protein, and *h_tot_* was calculated for wheat gluten at 8.3 mequiv/g protein. At pH 8.0, α was 0.790 at 37 °C [27].

### 4.3. Preparation of AFs

The AFs were prepared following a previous method reported by Lambrecht et al., with minor modification [24]. WGP powders with different DHs were reconstituted to a concentration of 2.5%, the pH value was adjusted to 7 with HCl (1 mol/L), and the incubation was continued for 48 h at 85 °C. The formed wheat gluten amyloid fibrils were then freeze-dried for subsequent analysis.

### 4.4. SDS-Polyacrylamide Gel Electrophoresis (SDS-PAGE)

Molecular weights of WGPs and AFs of DH4–8 were determined using a gel electrophoresis instrument (ChemiDoc MP; Bio-Rad Laboratories, Inc., Hercules, CA, USA). SDS-PAGE was carried out using a gel, including 15% separating gel and 4% stacking gel. The protein samples were mixed at the ratio of 10 µL of 5 × SDS-PAGE loading buffer (Biotopped, Beijing, China) for every 40 µL of sample, mixed, and then incubated in a thermostatic water bath at 100 °C for 180–300 s to denature the proteins. After cooling to room temperature, the samples were centrifuged for 4 min at 12,000× *g*, and the supernatant was used to load the gel. The current and electrophoresis time were 80 mA and 120 min, respectively. Coomassie Brilliant Blue R-250 was applied in the staining gel [4].

### 4.5. ThT Fluorescence Assay

AFs formation was monitored using ThT fluorescence as previously described [41]. The ThT powder was dissolved using phosphate-buffered solution to obtain 3 mM ThT reserve solution. The ThT reserve solution was filtered out of insoluble substances using a 0.22 μm syringe filter, and diluted 50-fold with phosphate buffer (pH 7, 10 mM) to become a working solution. We added 190 μL of samples to be tested and 10 μL of 0.6 mM ThT to a black 96-well plate. All ThT experiments were conducted in triplicate. All data represent the average of three independent measurements. Analysis was performed using the Synergy enzyme labeler (BioTek) with 440 nm excitation and 480 nm emission wavelengths. Measurements were taken hourly for a total of 48 h.

### 4.6. Surface Hydrophobicity Analysis

The surface hydrophobicity of WGPs and AFs with different DHs was determined using ANS (8 mM dissolved in 0.01 phosphate-buffered solution). Each sample was performed independently three times. All data represent the average of three independent measurements, and the error bars depict the standard deviation, and were calculated as previously described [42]. The protein sample was diluted to the gradient concentration (0.05–0.5 mg/mL) using phosphate-buffered solution (0.01 mol/L). Then, ANS solution (10 μL) and sample (200 μL) were added to the black 96-well plate. Each sample was tested three times in parallel. Fluorescence was analyzed using a Synergy Enzyme Labeler (BioTek Instruments, Inc., Winooski, VT, USA) at 390 nm excitation and 480 nm emission. Blank samples were 200 μL of PBS buffer solution. The relative fluorescence intensity was (sample—blank sample)/blank sample. The slope of the standard curves generated from the relative fluorescence intensity were used as the surface hydrophobicity index.

### 4.7. ζ-Potential Measurements

The ζ-potentials of WGPs and AFs with different DHs were measured by Zetasizer Nano ZS (Mastersizer 3000; Malvern Panalytical Ltd., Malvern, UK). WGPs and AFs with different DHs were diluted to 2 mg/mL with deionized water, and adjusted to pH 7 with HCl (1 M). Samples were added to capillary ζ-cells (Malvern Instruments) using a syringe, and three parallel experiments were performed for each sample with 20 tests. All data represent the average of three independent measurements, and the error bars depict the standard deviation.

### 4.8. FTIR Spectroscopy

FTIR spectroscopy and secondary structure analysis of WGPs and AFs powders with different DHs were performed using a Thermo Electron Nicolet iS10 FTIR spectrometer, using 64 scans with the range of 4000–400 cm^−1^ and resolution of 4 cm^−1^ to collect spectra, and background was collected before testing each sample [4]. The position and number of peaks were measured by the OMNIC program with the automatic peak finding function. Secondary structures were analyzed by Peak 4 1.8 software. All spectra for secondary structure analysis were taken from the wavelength range 1575–1715 cm^−1^ and were baseline-corrected and fitted using the Gaussian peak function. Then, the resulting curve was integrated to obtain the content of the secondary structure in the spectrum. The absorbance assignments table of the secondary structure is shown in Table 4 [13].

### 4.9. SEM

The freeze-dried AFs were mounted onto short aluminum SEM stubs. SEM photographs of AFs with different DHs were obtained using the Hitachi S4800 SEM. The accelerating voltage and working distance were 10 kV and 10 mm. The diameters of the AFs in the SEM images were analyzed using Image J software. The diameter of AF was measured every 1 μm for 3 times in total, and the average value was taken as the diameter of the AF for each DH.

### 4.10. TEM

The samples were negatively stained using uranyl acetate. The morphology structures of DH4–8 AFs were observed by TEM. The sample concentration was diluted to 0.1% (W_protein_/V) and examined by the JEM-1400plus instrument at 120 kV [37].

### 4.11. Statistical Analysis

Significant differences (*p* < 0.05) based on independent triplicate tests were determined with a one-way ANOVA procedure using SPSS Statistics 20.0. *t*-tests were performed using GraphPad Prism 8.0.1 to determine significant differences (* *p* < 0.05, ** *p* < 0.01, *** *p* < 0.001, **** *p* < 0.0001). Data processing was performed in OriginPro 9.8. software 2021.

## Figures and Tables

**Figure 1 ijms-24-13529-f001:**
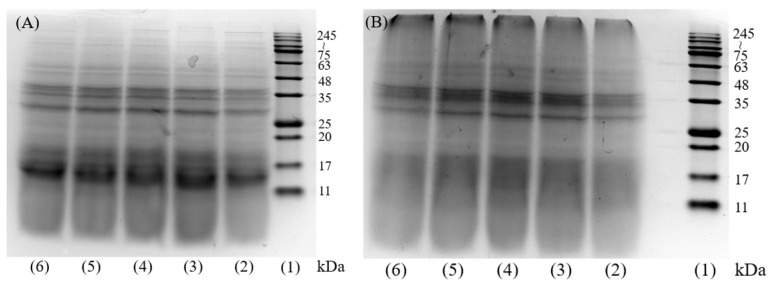
SDS-PAGE image of DH4–8 WGPs and AFs. (**A**): (1) molecular weight marker, (2) DH4 WGPs, (3) DH5 WGPs, (4) DH6 WGPs, (5) DH7 WGPs, (6) DH8 WGPs. (**B**): (1) molecular weight marker, (2) DH4 AFs, (3) DH5 AFs, (4) DH6 AFs, (5) DH7 AFs, (6) DH8 AFs.

**Figure 2 ijms-24-13529-f002:**
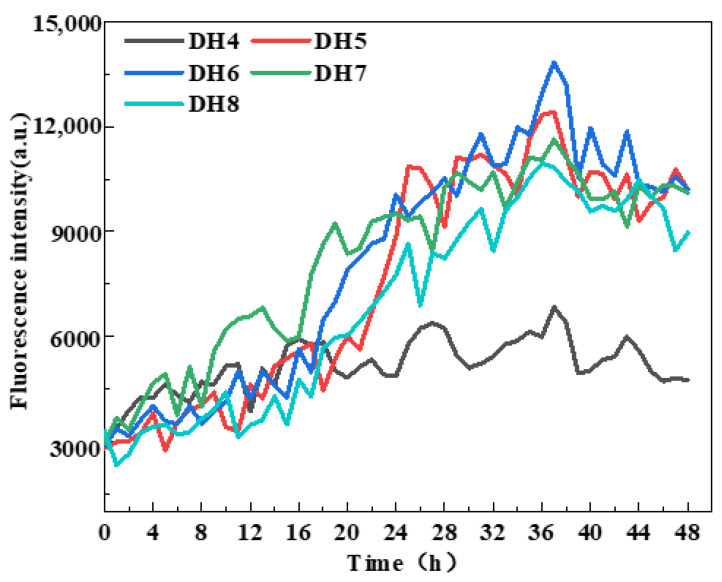
ThT fluorescence monitoring during AF formation with five different hydrolysis degrees.

**Figure 3 ijms-24-13529-f003:**
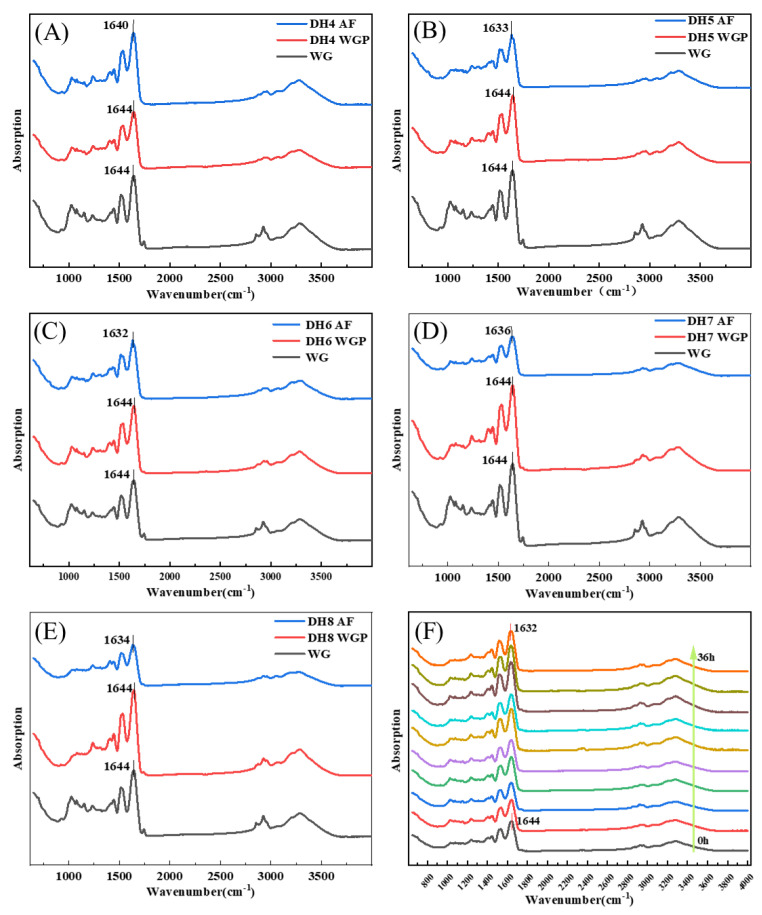
FTIR spectra of different degrees of hydrolysis from WG to WGP to AF. (**C**) From WG to DH6 AF, the peak of the amide I band moved from 1644 cm^−1^ to 1632 cm^−1^. The changes of peaks in the amide I band of other DHs were (**A**) 1644 cm^−1^–1640 cm^−1^ (DH4), (**B**) 1644 cm^−1^–1633 cm^−1^ (DH5), (**D**) 1644 cm^−1^–1636 cm^−1^ (DH7), and (**E**) 1644 cm^−1^–1634 cm^−1^ (DH8), respectively. (**F**) FTIR spectra show the AF formation process of DH6 WGPs, which was measured every 4 h. The arrow indicates from 0 h to 36 h.

**Figure 4 ijms-24-13529-f004:**
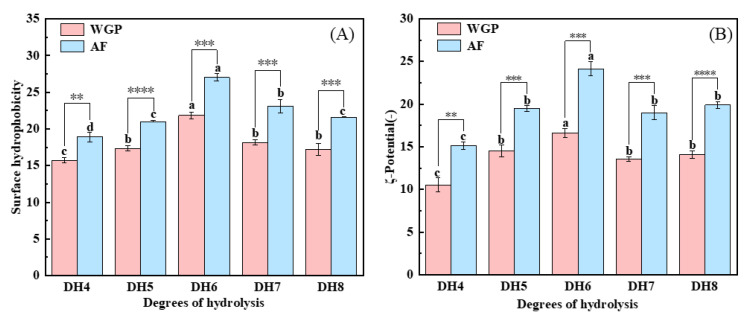
(**A**) Surface hydrophobicity analysis and (**B**) ζ-potential analysis of DH4–8 WGPs, DH4–8 AFs. Different letters of the different DH indicate significant differences (*p* < 0.05). **, ***, **** represents significant difference between two sets of data with the same DH (** *p* < 0.01, *** *p* < 0.001, **** *p* < 0.0001).

**Figure 5 ijms-24-13529-f005:**
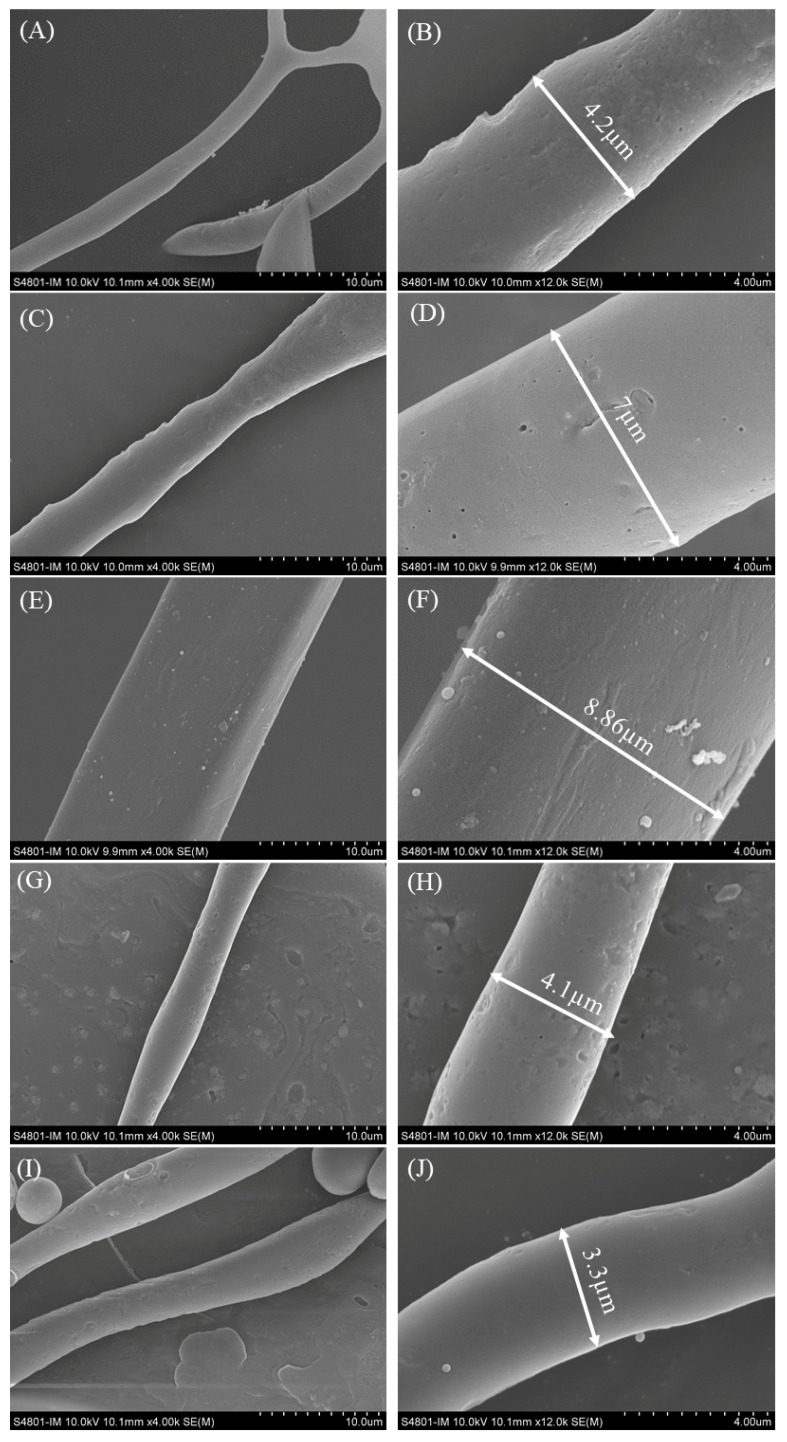
SEM images of DH4–8 AFs at 4 k and 12 k times. Image was taken at 4000 times (10 µm) and 12,000 times (4 µm) for each degree of hydrolysis AFs. (**A**) DH4 AF × 4 k; (**B**) DH4 AF × 12 k; (**C**) DH5 AF × 4 k; (**D**) DH5 AF × 12 k; (**E**) DH6 AF × 4 k; (**F**) DH6 AF × 12 k; (**G**) DH7 AF × 4 k; (**H**) DH7 AF × 12 k; (**I**) DH8 AF × 4 k; (**J**) DH8 AF × 12 k.

**Figure 6 ijms-24-13529-f006:**
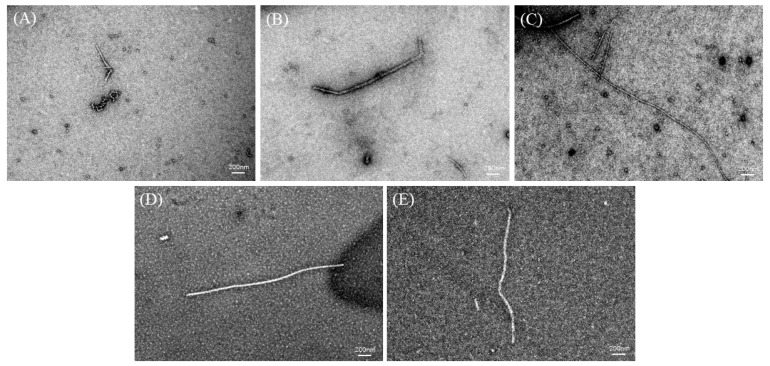
TEM images of DH4–8. The white line in the images is the scale bar: 200 nm. (**A**) DH4 AF; (**B**) DH5 AF; (**C**) DH6 AF; (**D**) DH7 AF; (**E**) DH8 AF.

**Figure 7 ijms-24-13529-f007:**
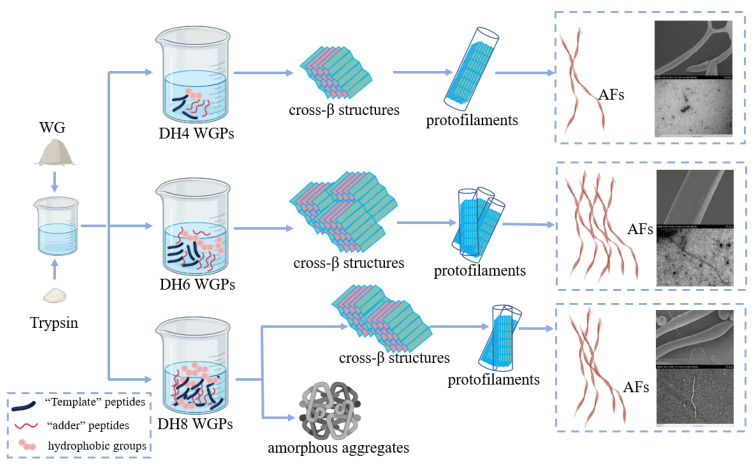
The mechanisms of different DH-regulated WG fibrillations (taking DH4, DH6, and DH8 as examples).

**Table 1 ijms-24-13529-t001:** Hydrolysis time with different hydrolysis degrees and different enzyme–substrate ratios ^a^.

Degrees of Hydrolysis	Enzyme–Substrate Ratio		
	1:100	1:500	1:1000
DH4	1.3 ± 0.1 (h) ^a^	2.2 ± 0.2 (h) ^a^	4.3 ± 0.2 (h) ^b^
DH5	2.7 ± 0.15 (h) ^a^	3.5 ± 0.15 (h) ^a^	5.8 ± 0.32 (h) ^b^
DH6	4.5 ± 0.25 (h) ^a^	4.5 ± 0.18 (h) ^a^	7 ± 0.31 (h) ^b^
DH7	6.1 ± 0.25 (h) ^a^	7.5 ± 0.25 (h) ^a^	10 ± 0.25 (h) ^b^
DH8	7.6 ± 0.15 (h) ^a^	8.5 ± 0.15 (h) ^a^	12 ± 0.31 (h) ^b^

^a^ Different superscript letters in the same line indicate significant differences (*p* < 0.05). The values in parentheses indicate standard deviation.

**Table 2 ijms-24-13529-t002:** Proportion of protein secondary structure of WGPs in different DHs.

Degrees of Hydrolysis	Area (%)				
	β-Sheet	Random Coil	α-Helix	Antiparallel β-Sheet	β-Turn
DH4	30.7484	17.3761	34.5704	9.0298	12.2753
DH5	31.9271	15.7266	33.3201	8.067	10.9592
DH6	31.3268	14.8518	34.9059	8.1665	10.7489
DH7	30.1374	16.1288	34.1976	8.2575	11.2787
DH8	29.7842	17.635	31.6121	8.8613	12.1073

**Table 3 ijms-24-13529-t003:** Proportion of protein secondary structure of AFs in different DHs.

Degrees of Hydrolysis	Area (%)					
	Δβ-Sheet *	Β-Sheet	Random Coil	α-Helix	Antiparallel β-Sheet	β-Turn
DH4	13.037	43.7854	25.1712	14.3445	5.6842	10.6739
DH5	14.3558	46.2829	16.7061	17.7164	6.0611	12.9063
DH6	19.1468	50.4736	21.5308	12.6679	5.7047	9.48678
DH7	14.9632	45.1006	28.0488	10.3465	5.7071	10.5248
DH8	14.6054	44.3896	15.5662	23.7324	6.1335	10.1332

* Δβ-sheet is the β-sheet content of AFs with different degrees of hydrolysis minus the β-sheet content of WGPs.

**Table 4 ijms-24-13529-t004:** Assignment of amide I absorbances.

Amide I Structural Assignment	Wavenumber (cm^−1^)
cross-β	1611–1637
Random coil	1637–1647
α-helix	1647–1662
β-turn	1662–1678; 1689–1699
antiparallel β-sheet	1679–1688

## Data Availability

All data needed to evaluate the conclusions of the paper are provided in the paper or as Appendix A. The datasets generated and/or analyzed in this study are available from the corresponding author upon reasonable request.

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
