# Peer review of "Degree of Hydrolysis Regulated by Enzyme Mediation of Wheat Gluten Fibrillation: Structural Characterization and Analysis of the Mechanism of Action"

_ijms, 2023, doi:10.3390/ijms241713529_

Round 1

Reviewer 1 Report

In this study authors explore the impact of different degrees of hydrolysis (DH) on fibrillation of wheat gluten peptides (WGPs) mediated by trypsin. Overall, the manuscript contains interesting experimental findings, but it requires detailed data presentation, and a more comprehensive discussion of the results and their interpretations to enhance the clarity and readability of the paper. 

I have following concerns that need to be addressed:

1.     The manuscript lacks details on the number of replicates for each experiment and does not mention any statistical analysis to validate the results.

2.     The manuscript provides information about the enzymatic time and optimal enzyme-substrate ratio for different degrees of hydrolysis DH. The authors should provide a more clear explanation of how the enzymatic time relates to the degree of hydrolysis.

3.     Line 161: The manuscript mentions that the findings are similar to those of Ridgley et al., but there is no explanation of what those findings were. Similarly, the manuscript mentions the work of other researchers, but it does not provide sufficient context or explanation of how the current findings relate to those studies. It is essential to include a comparative discussion of the similarities and differences between the present results and the findings of previous research to demonstrate the novelty and relevance of the current work. 

4.     The full details of SEM and TEM analysis is missing, such as the number of AFs selected for measurement and any potential sources of error or uncertainty in the diameter or morphology determination of the AFs. Additionally, a scale bar is missing in Figure 6."

5.     The proposed mechanism relies on the different degrees of hydrolysis DH of WG influencing the formation of AFs. However, the manuscript lacks quantitative data on the concentration of 'template' and 'adder' peptides at different DH levels. Providing quantitative analysis and data on peptide concentrations would strengthen the mechanism's validity.

6.     The authors should elaborate on the potential applications of their results and how they contribute to the understanding of WGPs fibrillation and AFs formation and its industrial implications.

Reviewer 2 Report

The mauscript "Degree of Hydrolysis-Regulated by Enzyme-mediated on Wheat Gluten Fibrillation: Structural Characterisation and Analysis of the Mechanism of Action" has an actual subject of the research field.

This is an important direction to expand the application of wheat gluten is based on trypsin mediated wheat gluten fibrillation.

The optimal conditions in terms of pH, temperature and protein concentration have been investigated, although there have been some studies using trypsin to enzymatically digest wheat gluten proteins to produce amyloid protofibrils.

Fluorescence detection, FTIR absorption spectroscopy, SEM and TEM showed that the hydrolysis of WGs to DH6 could produce a higher number of AFs with better morphology, the highest surface hydrophobicity and a more stable system under suitable conditions.

The manuscript has a good general presentation.

The procedures and methods were adequate.

Results are interesting and useful.

The conclusions were based o the obtained results

The references can be upgraded (some valuable reference were lost)!

Round 2

Reviewer 1 Report

I believe that the authors have properly addressed the main concerns that I had and hence I now do support the publication.